# Conditioning Convolutional Segmentation Architectures with Non-Imaging Data

**Grzegorz Jacenków**[1]                                        G.JACENKOW@ED.AC.UK
**Agisilaos Chartsias**[1]
**Brian Mohr**[2]
**Sotirios A. Tsaftaris**[1,3]

[1] *The University of Edinburgh, Edinburgh, United Kingdom*

[2] *Canon Medical Research Europe Ltd., Edinburgh, United Kingdom*

[3] *The Alan Turing Institute, London, United Kingdom*

## Abstract

We compare two conditioning mechanisms based on concatenation and feature-wise modulation to integrate non-imaging information into convolutional neural networks for segmentation of anatomical structures. As a proof-of-concept we provide the distribution of class labels obtained from ground truth masks to ensure strong correlation between the conditioning data and the segmentation maps. We evaluate the methods on the ACDC dataset, and show that conditioning with non-imaging data improves performance of the segmentation networks. We observed conditioning the U-Net architectures was challenging, where no method gave significant improvement. However, the same architecture without skip connections outperforms the baseline with feature-wise modulation, and the relative performance increases as the training size decreases.

**Keywords:** Segmentation, Cardiac MRI, Side Information, Convolutional Neural Network.

## 1. Introduction

Integrating non-imaging modalities becomes of interest to the research community with dedicated workshops such as beyondMIC 2018. The majority of the work has been focused on multi-modal data fusion where each modality is mapped to an embedding otherwise by concatenating (Tiwari et al., 2011) or maximising correlation between the views (Golugula et al., 2011). Other approaches include intermediate functions such as neural networks combined with a linear classifier. For instance, (Shmulev and Belyaev, 2018) developed a method to predict conversion of mild cognitive impairment to Alzheimer's disease, and (Cerna et al., 2019) shown a classifier to estimate probability of mortality within one year. However, to the best of our knowledge the current techniques do not apply to the segmentation networks as they focus on classification.

In this work, we use segmentation of Cardiovascular Magnetic Resonance (CMR) images as an example for evaluating the conditioning mechanisms. We observe that the majority of the proposed approaches for CMR segmentation rely only on imaging data, and do not incorporate additional information available in Electronic Health Records (EHRs) (Bizopoulos and Koutsouris, 2019). As collecting such information is often more time-efficient than the

annotation process, we motivate to investigate how non-imaging data can be used as prior in convolutional segmentation networks. To ensure strong correlation of the conditioning information with the segmentation task, and to avoid inter-subject bias due to pathologies, we propose a proof-of-concept where the network is conditioned on the distribution of class labels obtained from the ground truth masks. Together with the image, we provide to the networks expected percentage of pixels in the output mask corresponding to each class, i.e. myocardiun, left- and right ventricular cavities. This conditioning data is an approximation of the heart's size, which is a common biomarker easily extracted from echocardiography images (Jenkins et al., 2008).

## 2. Methodology

We consider concatenation-based conditioning and feature-wise modulation applied to two networks for 2D segmentation; a U-Net (Ronneberger et al., 2015) where each convolutional layer is followed by batch normalisation (Ioffe and Szegedy, 2015), and an encoder-decoder that has the same architecture as the U-Net except there are no skip connections.

**Concatenation-based conditioning** refers to methods where the conditioning information is concatenated with a feature map or with the model's input. We evaluate two approaches for acquiring the conditioning representation $\tilde{z}$. Given a distribution of class labels $z$ [1] and a function $f$, $\tilde{z} = f(z)$, where $f$ is either an identity function (referred as *raw concatenation*) or a fully-connected 3-6-12-6-3 network (referred as *MLP concatenation*). We apply the concatenation-based conditioning at three levels: *early fusion* with spatial replication of the input-level features, *middle fusion* at the latent space of the encoder-decoder networks, and *late fusion* before the last convolutional layer.

**Feature-wise Linear Modulation** (FiLM) (Perez et al., 2018) can be classified as an instance normalisation (Ulyanov et al., 2016) technique in which a scaling $\gamma$ and a shifting $\beta$ factors are applied to a particular channel $c$ in a feature map $F_c$, i.e. $\text{FiLM}(F_c|\gamma_c, \beta_c) = \gamma_c F_c + \beta_c$. In contrast to the regular instance normalisation, the factors are learnt with a multilayer perceptron from an input $z$. Our work focuses on applying FiLM layers along the decoder path (*decoder fusion*) and before the final convolutional layer (*late fusion*).

## 3. Experiments

**Dataset.** We use the cardiac cine-MRI dataset from the ACDC 2017 challenge (Bernard et al., 2018) for the task of segmenting the images into three anatomical structures, i.e. myocardium, left- and right ventricular cavities. The annotated dataset contains images at end-systolic and -diastolic phases from 100 patients, and varying spatial resolutions. We resample the volumes to a common resolution of $1.37$ mm$^2$ per pixel, resize each slice to 224 x 244 pixels, and standardise intensities using z-score with clipping values exceeding three units of standard deviation from the volume's mean.

**Training and Evaluation.** All models are trained using Adam (Kingma and Ba, 2014) optimiser with learning rate $\alpha = 0.0001$, and Focal Loss (Lin et al., 2017) with $\gamma = 0.5$. The networks are trained with 500 epochs and we apply early stopping with patience set to 100. To determine the effect of conditioning mechanisms on datasets with limited amount

---

1. To address the class imbalance, we exclude background labels and multiple the other classes by 100.

| Fraction | Baseline | Concatenation (raw) | | | Concatenation (MLP) | | | FiLM | |
|---|---|---|---|---|---|---|---|---|---|
| | | Early | Middle | Late | Early | Middle | Late | Decoder | Late |
| 100% | $.89_{\pm.04}$ | $.89_{\pm.03}$ | $.90^*_{\pm.03}$ | $.90_{\pm.04}$ | $.89_{\pm.03}$ | $.90_{\pm.03}$ | $.90_{\pm.03}$ | $\mathbf{.91}^*_{\pm.02}$ | $.90^*_{\pm.03}$ |
| 25% | $.80_{\pm.13}$ | $.81_{\pm.10}$ | $.82_{\pm.09}$ | $\mathbf{.83}^*_{\pm.10}$ | $.81_{\pm.11}$ | $.82_{\pm.10}$ | $.82_{\pm.10}$ | $.81_{\pm.12}$ | $.82_{\pm.10}$ |
| 6% | $.39_{\pm.29}$ | $.53^*_{\pm.23}$ | $.58^*_{\pm.27}$ | $\mathbf{.59}^*_{\pm.25}$ | $.58^*_{\pm.28}$ | $.54^*_{\pm.26}$ | $.58^*_{\pm.25}$ | $.55^*_{\pm.26}$ | $.55^*_{\pm.26}$ |
| 1.5% | $.41_{\pm.23}$ | $.42_{\pm.23}$ | $\mathbf{.44}_{\pm.22}$ | $.42_{\pm.22}$ | $\mathbf{.44}_{\pm.24}$ | $.42_{\pm.22}$ | $.42_{\pm.22}$ | $.38_{\pm.24}$ | $.34^*_{\pm.23}$ |

Table 1: Performance of the networks with U-Net architecture as an average over Dice scores for LVC, myocardium and RVC. The best results are shown in **bold**. An asterisk (*) denotes the statistical significance (5%) comparing to the baseline.

| Fraction | Baseline | Concatenation (raw) | | | Concatenation (MLP) | | | FiLM | |
|---|---|---|---|---|---|---|---|---|---|
| | | Early | Middle | Late | Early | Middle | Late | Decoder | Late |
| 100% | $.87_{\pm.04}$ | $.86^*_{\pm.04}$ | $.88^*_{\pm.03}$ | $.88^*_{\pm.03}$ | $.87_{\pm.04}$ | $.88^*_{\pm.03}$ | $.87^*_{\pm.03}$ | $\mathbf{.89}^*_{\pm.02}$ | $.88^*_{\pm.03}$ |
| 25% | $.78_{\pm.09}$ | $.75^*_{\pm.11}$ | $.75^*_{\pm.12}$ | $.78_{\pm.09}$ | $.78_{\pm.10}$ | $.76_{\pm.11}$ | $.77_{\pm.10}$ | $\mathbf{.82}^*_{\pm.06}$ | $.78_{\pm.10}$ |
| 6% | $.55_{\pm.22}$ | $.53_{\pm.22}$ | $.53_{\pm.22}$ | $.55_{\pm.23}$ | $.52^*_{\pm.23}$ | $.51^*_{\pm.23}$ | $.54_{\pm.22}$ | $\mathbf{.58}_{\pm.19}$ | $.48^*_{\pm.24}$ |
| 1.5% | $.31_{\pm.17}$ | $.34_{\pm.19}$ | $.31_{\pm.17}$ | $.31_{\pm.19}$ | $\mathbf{.38}^*_{\pm.21}$ | $.29_{\pm.18}$ | $.35^*_{\pm.19}$ | $.37^*_{\pm.18}$ | $.31_{\pm.18}$ |

Table 2: Same as Table 1 but for the encoder-decoder architecture.

of training examples, we repeat each experiment with varying fractions of the training set, i.e. at 100%, 25%, 6% and 1.5% (single subject). The experiments are evaluated using 3-fold cross validation with subjects shuffled and split into 70%, 15%, 15% training, validation and test sets respectively. We calculate a Dice score for 3D volumes as the average across all anatomical structures. We report average Dice, standard deviation and evaluate statistical significance (5%) using paired t-test with Bonferroni correction on the test sets as we compare each conditioning mechanism with the corresponding baseline, i.e. we make 8 comparisons.

**Results.** The empirical results of the U-Net and the encoder-decoder networks are presented in Table 1 and Table 2 respectively. Overall, it can be seen that conditioning on non-imaging information improves segmentation performance in terms of Dice. In particular future-wise modulation has relative improvement of 2% - 19% over the encoder-decoder baseline. We also observe that relative performance increases as the size of the training dataset decreases. However, the U-Net architectures show to be more challenging for integrating non-imaging information. Although, concatenation of raw values before the last convolutional layer has outperformed the U-Net baseline by a margin of 1% - 51%, the variance remains high. Furthermore, the results do not show significant relative improvement with limited training examples as in the encoder-decoder networks.

## 4. Conclusion

We have considered the task of conditioning segmentation networks on non-imaging data. We have shown that conditioning with non-imaging data improves performance of the segmentation networks with feature-wise modulation for the encoder-decoder networks yielding a consistent improvement. However, conditioning the U-Net networks is challenging and the same methods do not result in significant improvement.

## Acknowledgments

This work was supported by the Engineering and Physical Sciences Research Council [grant number EP/R513209/1]; and Canon Medical Research Europe Ltd. S.A. Tsaftaris acknowledges the support of the Royal Academy of Engineering and the Research Chairs and Senior Research Fellowships scheme.

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
