# OpenReview forum: "Conditioning Convolutional Segmentation Architectures with Non-Imaging Data"
_MIDL.io/2019/Conference/Abstract — MIDL Abstract 2019_

### Official Review · AnonReviewer1 · 2019-04-30
**Interesting abstract**

**Rating:** 3
**Confidence:** 3

**Review:**

This paper investigates the task of conditioning segmentation networks with non-imaging data. This is an interesting paper, but reading through the paper, it is hard to figure out what type of non-imaging data the authors refer to!? Multimodal (imaging and non-imaging) analysis has a large history in medical imaging, but the paper is written in a way that ignores all of them.

---

### Official Review · AnonReviewer2 · 2019-05-01
**.**

**Rating:** 2
**Confidence:** 3

**Review:**

Uses non-imaging data, in the form of histograms (distribution of class labels obtained from the ground truth masks), to improve the segmentation. This is an important problem. However, the results show limited promise. Details of the network architecture are absent.

---

### Decision · Program_Chairs · 2019-05-06
**Acceptance Decision**

Accept